# Systemic Innate Immune System Restoration as a Therapeutic Approach for Neurodegenerative Disease: Effects of NP001 on Amyotrophic Lateral Sclerosis (ALS) Progression

**DOI:** 10.3390/biomedicines12102362

**Published:** 2024-10-16

**Authors:** Michael S. McGrath, Rongzhen Zhang, Paige M. Bracci, Ari Azhir, Bruce D. Forrest

**Affiliations:** 1Department of Medicine, University of California San Francisco, San Francisco, CA 94110, USA; 2Neuvivo, Inc., Palo Alto, CA 94301, USA; 3Department of Epidemiology and Biostatistics, University of California San Francisco, San Francisco, CA 94158, USA; 4Hudson Innovations, LLC, Nyack, NY 10960, USA

**Keywords:** ALS, innate immune activation, macrophage, vital capacity, creatinine, survival, NP001

## Abstract

Background/objective: Amyotrophic lateral sclerosis (ALS) is a diagnosis that incorporates a heterogeneous set of neurodegenerative processes into a single progressive and uniformly fatal disease making the development of a uniformly applicable therapeutic difficult. Recent multinational ALS natural history incidence studies have identified systemic chronic activation of the innate immune system as a major risk factor for developing ALS. Persistent immune activation in patients with ALS leads to loss of muscle and lowering of serum creatinine. The goal of the current study was to test whether the slowing of nerve and muscle destruction in NP001-treated ALS patients compared with controls in phase 2 studies would lead to extension of survival. Methods: Phase 2 clinical studies with NP001, an intravenously administered form of the innate immune system regulator NaClO_2_, are now reporting long-term survival benefits for drug recipients vs. placebo controls after only six months of intermittent treatment. As a prodrug, NP001 is converted by macrophages to taurine chloramine, a long-lived regulator of inflammation. We performed a pooled analysis of all patients who had completed the studies in two six-month NP001 phase 2 trials. Changes in respiratory vital capacity and the muscle mass product, creatinine, defined treated patients who, compared to placebo, had up to a year of extended survival. Conclusions: The observed longer survival in ALS patients with the greatest inflammation-associated muscle loss provides further evidence that ALS is a disease of ongoing innate immune dysfunction and that NP001 is a disease-modifying drug with sustained clinical activity.

## 1. Introduction

The pathogenesis of ALS is heterogeneous, and the genetic abnormalities identified in small subsets of ALS patients have failed to define a comprehensive approach to the disease [1,2]. The finding of misfolded proteins and pathogenic aggregation of those proteins have also failed to define a therapeutically validated approach to ALS [3]. Many studies have suggested a role for inflammation, but, to date, there are no disease-specific biomarkers identified nor anti-inflammatory approaches that have been successful in the treatment of ALS [4,5].

A recent natural history study of ALS pathogenesis from Sweden tracked levels of creatinine with a parallel evaluation of an innate immune system acute phase protein (APP), C-reactive protein (CRP), in 525 ALS patients compared with 2650 age-matched controls [6]. Creatinine blood levels began to drop before ALS symptom onset with nadir at a time after diagnosis when CRP blood levels were highest. Normally, the innate immune acute phase reaction is self-limited; however, these observations suggest that a component of ALS is a chronic, acute phase response, and this process is best demonstrated by the low creatinine levels with CRP elevation tracking with disease activity [7]. The severity of muscle mass loss in ALS patients parallels this process [7,8,9]. A meta-analysis of a series of natural history studies [8] showed that patterns of innate immune activation in these natural history studies were best defined in ALS patients ≤ 65 years of age as older patient immune activation pattern complexities confounded this disease association [6,8].

These natural history studies provide a mechanistic linkage between CRP, an APP reactant, and creatinine, a byproduct of muscle metabolism. Reeds et al. [9] provide evidence for a mechanism by which the acute phase response requires excessive consumption of muscle as a source of relatively rare amino acids present in the APPs (phenylalanine, tryptophan, tyrosine). Thus, there is a direct linkage between an ongoing innate immune response, the lowering of creatinine, and the progression of muscle loss that characterizes ALS. When combined, these data reinforce the existence of a subset of ALS that is defined by innate immune activation with elevation of CRP and lowering of serum creatinine in patients up to 65 years of age [6,7,8].

NP001 is a stabilized form of sodium chlorite for intravenous use. It has been studied in three clinical trials in ALS patients. Phase 1 studies [10] showed a dose-dependent downregulation of monocyte activation markers after a single dose, measured 24 h later. In the first phase 2 study, NP001 showed a dose-dependent slowing of the ALS functional rating scale (ALSFRS-R) measured loss of function, and that activity was in patients with evidence for innate immune activation as measured by elevated plasma CRP levels [11]. The ALSFRS-R is a 0–48-unit scale that measures 12 functional categories, assigning 0–4 units/function. This is the usual scale for following ALS disease in patients, losing an average of 0.9 units of function per month [11]. A recent post hoc analysis of both phase 2 six-month clinical trials confirmed clinical activity in ALS patients fitting the description of those with innate immune activation. Patients ≤ 65 having an elevated level of plasma CRP had the best clinical outcome as measured by slowing loss of ALSFRS-R and VC scores over 6 months compared to placebo [12].

The loss of muscle in ALS patients is manifest as sarcopenia in the skeletal and respiratory function muscles, leading to loss of vital capacity, a function linked to overall survival [13]. Both forced vital capacity (FVC) and slow vital capacity (SVC) measures have been employed to monitor the respiratory function of ALS patients. In the most complete published evaluation of these tests, Andrews et al. tracked both measures in relation to disease progression rates in over 3500 ALS patients involved as placebos in clinical trials. On average, patients ≤ 65 years of age lost approximately 2.5% of their VC/month; results for ALS patients > 65 years showed greater variation in VC function with an average VC loss of 3.6%/month [13]. In evaluating clinically significant vital capacity rates of progression, they noted that cutting the rate of VC loss by half prolonged life. Thus, innate immune dysfunction that allows or promotes muscle loss can be assessed quantitatively by evaluation of creatinine and VC changes from baseline and the relation of those changes to the duration of overall survival.

Here, we propose that a treatment targeting the innate immune dysfunction of ALS, e.g., NP001, can be assessed quantitatively by evaluation of creatinine and VC changes from baseline, and these changes will be related to overall survival (OS) for ALS patients. Figure 1 illustrates this novel theory of ALS pathogenesis as being an immune activation disorder with ALS initiation at the neuromuscular junction (NMJ) driving the pathogenesis of disease. To test this hypothesis, we used data collected from patients with ALS who participated in two six-month phase 2 studies of NP001 [12,14] that were recently reported in a follow-up study by Forrest et al. [15], which showed improved OS in those patients treated with NP001.

## 2. Materials and Methods

### 2.1. Clinical Trials Overview

Two phase 2 trials were conducted by Neuraltus (Palo Alto, CA, USA) in ALS patients (NCT01281631 and NCT02794857 as registered at ClinicalTrials.gov). These were both placebo-controlled 6-month studies. Phase 2A was completed in 2012 [14], and phase 2B in 2017 [12]. No drug-related Serious Adverse Events (SAEs) occurred in either Phase 2A or 2B trials [12,14].

Both phase 2A and 2B studies were approved by the clinical site institutional ethics committees, and informed consent was obtained from all participants.

### 2.2. Description of ALS Phase 2A and 2B Trials and Participants

Details of these two six-month trials have been published [12,14]. For purposes of the current combined phase 2A + 2B studies, only patients treated with 2 mg/kg of NP001 chlorite or placebo were included. The dosage of NP001 used in both clinical trials is 2 mg/kg body weight as chlorite (equivalent to 2.682 mg/kg sodium chlorite).

In both studies, all patients were enrolled within 3 years of symptom onset. The additional criteria for all patients enrolled in the phase 2B study included a plasma high-sensitivity C-reactive protein (hs-CRP) concentration of >1.13 mg/L at the pre-screening visit. The units expressed throughout the paper are hs-CRP units, abbreviated as CRP. For each trial, patients were planned to receive a total of 20 infusions administered intravenously over 6 cycles during a 6-month study, with 4 weeks between the start of each cycle. Cycle 1 consisted of 30-minute infusions over 5 consecutive days. Cycles 2, 3, 4, 5, and 6 each consisted of 3 consecutive 30-minute daily infusions.

### 2.3. Analysis of Clinical Outcome Data

The current study focused on the percentage change in predicted vital capacity (VC) over the six-month studies. The phase 2A trial assessed the predicted VC in FVC, whereas phase 2B assessed predicted the VC in SVC. Since predicted VC values between FVC and SVC are comparable [16], the evaluation of NP001 effects on VC combined changes from both trials normalized to “% VC change from baseline [100 × (predicted VC at study end − predicted VC at baseline)/predicted VC at baseline]”.

Analyses presented here are limited to the completers, defined as all patients who received NP001 chlorite at a dose of 2 mg/kg or placebo and had completed the 6-cycle treatments with ALSFRS-R total score assessment at the end of the study, resulting in a total of 189 participants, 91 of NP001 2 mg/kg chlorite and 98 of placebo, analyzed.

### 2.4. Serum Creatinine Levels in Phase 2A and Phase 2B

The serum levels of creatinine as safety biomarkers were collected at baseline/pre-screening in both phase 2A and 2B trials and grouped for analyses based on clinical threshold levels. Given sex differences in muscle mass [17,18], baseline creatinine was further categorized by accounting for sex. The participants with serum creatinine at baseline for males < 71 µM/L or females < 53 µM/L were defined as the low creatinine group and the high creatinine group, which included those with serum creatinine ≥ 71 µM/L for males or ≥53 µM/L for females at baseline.

### 2.5. Phase 2A and 2B Completers Survival Analyses

Survival data for ALS patients in the intention-to-treat (ITT) population were collected and recently reported by Forrest et al. [15]. These data were obtained from Neuvivo Inc. Overall survival (OS) was defined as the time in months from date of randomization to date of death due to any cause or to last contact/known alive for patients lost to follow-up (censored) and were assessed through 30 September 2022.

### 2.6. Statistical Analyses

Statistical analysis was performed using JMP Pro 17 (SAS Institute, Cary, NC, USA) and SAS 9.4 (SAS Institute, Cary, NC, USA) and packages “survival 3.7-0” [19] and “survminer 0.4.9” [20] in R 4.4.1 [21]. Data were summarized as counts and percentages for categorical data and using standard univariate descriptive statistics (number of participants, mean, standard deviation, median) for continuous/discrete data by treatment group. Preliminary analyses of categorical data were analyzed using Fisher’s exact test and Chi-square tests, and continuous/discrete data were analyzed using *t*-tests or Wilcoxon rank sum tests as appropriate.

Overall survival differences by treatment/group were assessed using a log-rank test. Kaplan–Meier methods were used to estimate survival probabilities and curves. Cox-proportional hazards models were used to estimate the hazard ratio associated with dying for patients treated with NP001 compared with those given placebo. The assumptions of proportionality of hazards over time were assessed using the Schoenfeld test. Patients who were lost to follow-up or alive at the end of the follow-up period were considered censored in all survival analyses.

All statistical tests were two-sided and considered statistically significant for *p*-value < 0.05.

## 3. Results

### 3.1. Demographics and Clinical Characteristics

Patient demographics and disease characteristics did not differ by treatment groups in the completers, with the exception of a lower percentage of patients with familial ALS (96.7% sporadic) in the NP001 treatment group compared to the placebo group (82.7% sporadic) (Appendix A).

When familial patients were removed from any analysis, the outcomes of all group analyses were not statistically different. For the current study, survival analysis was performed on completers who had inflammation as documented by having a plasma hs-CRP level > 1.13 mg/L at baseline. Considering the disproportion in the ALS subtype, there had been many more familial cases assigned to placebo; placebos with and without familial patients included had survival curves indistinguishable from one another (Appendix A).

Table 1 shows patient demographics and disease characteristics of phase 2A and 2B completers with plasma CRP > 1.13 mg/L at baseline.

### 3.2. OS and Percent of VC Change

In the ALS patients who completed the 6-month studies (completers), the median OS was 38.6 months and 28.4 months in the 2 mg/kg NP001 chlorite and placebo groups, respectively (log-rank, *p* = 0.04). Patients treated with NP001 had a better survival outcome than those on placebo (hazard ratio (HR) = 0.72, 95% CI: 0.52, 0.99), as shown in Figure 2A.

In completers with plasma CRP > 1.13 mg/L at baseline, NP001 treatment was statistically significantly associated with survival duration. Median survival was 42.8 months and 28.4 months in the 2 mg/kg NP001 and placebo groups, respectively (Figure 2B, log-rank, *p* = 0.002; HR = 0.57, 95% CI: 0.39, 0.82 for NP001 vs. placebo group).

Figure 2C shows the % VC change from baseline of the NP001 treatment arm vs. placebo group in completers with plasma CRP > 1.13mg/L at baseline. The NP001 treatment arm lost 29% less respiratory function than the placebo arm by the end of the study (NP001 = −9.1% vs. placebo = −12.9%) (Wilcoxon, *p* = 0.02).

### 3.3. Effect of Baseline Serum Creatinine Level on Respiratory Function and OS

#### 3.3.1. Serum Creatinine Level by Patient Demographics

Figure 3 shows the baseline creatinine distributions in the low and high creatinine groups in phase 2A and 2B completers with baseline plasma CRP > 1.13 mg/L. The median serum creatinine level was statistically significantly different between the groups (53 µM/L vs. 71 µM/L in low vs. high groups, respectively (Wilcoxon, *p* < 0.0001).

Table 2 shows the baseline demographic and clinical characteristics of the creatinine group in phase 2A and 2B completers with baseline plasma CRP > 1.13 mg/L. ALS patients in the low creatinine group were significantly younger (Wilcoxon, *p* = 0.003) and had more advanced ALS disease, as defined by significantly lower ALSFRS-R score (Wilcoxon, *p* = 0.008) and longer duration since ALS symptom onset (Wilcoxon, *p* = 0.004).

#### 3.3.2. Baseline Serum Creatinine and Plasma CRP Values Define a Subset of NP001-Treated ALS Patients Whose Loss in VC over the 6-Month Study Is Markedly Slowed vs. Placebo

Further analyses were conducted to examine whether change in VC over time was associated with NP001 treatment related to baseline creatinine level. Figure 4A shows that loss of VC was slower over time in NP001-treated patients as compared to placebo in the low creatinine group. Those patients with low creatinine who were on placebo lost an average of 3.0% VC per month, whereas patients who were treated with NP001 lost 1.6% VC per month; a > 46% slower rate of VC loss among those treated with NP001 compared with placebo (Wilcoxon, *p* = 0.02). Note the change in VC over time did not differ by treatment status in participants with high creatinine levels at baseline (Figure 4B, Wilcoxon, *p* = 0.30).

No significant differences in baseline demographic and clinical characteristics were observed by the two treatment groups when analyses were restricted to the low creatinine group (Table 3) or the high creatinine group (Appendix A).

Overall survival analyses of patients with ALS were stratified by the baseline creatinine group (Figure 5). In analyses restricted to completers in the low creatinine group with baseline plasma CRP > 1.13 mg/L, the median survival was 45.5 months and 28.4 months in the 2 mg/kg NP001 and placebo groups, respectively (Figure 5A, log-rank, *p* = 0.005). Patients with ALS who were treated with NP001 were 40% as likely to have died compared with those on placebo (HR = 0.41, 95% CI: 0.22, 0.78). For patients in the high creatinine group (Figure 5B), survival duration did not differ by treatment group; median survival was 38.6 months and 29.3 months in the NP001 and placebo arms, respectively (Figure 5B, log-rank *p* = 0.17; HR = 0.73, 95% CI: 0.46, 1.1 for NP001 vs. placebo group).

## 4. Discussion

### 4.1. NP001 Clinical Studies in ALS Support a Mechanism of Action That Predicts Long-Term Effects

The data reported in this study utilized a combined database from two NP001 phase 2 placebo-controlled trials in patients with ALS. This analysis focused on the two measures directly related to muscle integrity: serum creatinine and VC. All creatinine and VC data from the combined studies were reported using a dataset including all patients who had completed the six-month trials with ALSFRS-R total score assessment at the end of the study. Because creatinine levels were an indirect measure of muscle loss, the ALS subset with hs-CRP plasma levels > 1.13 mg/L was used to test whether production of CRP might best be associated with changed creatinine levels and VC changes over time. The data presented link concordant changes in muscle-related measures of creatinine and VC over time with long-term survival after just a six-month trial. Notably, in the subset of ALS defined as having innate immune dysfunction (elevated CRP), the median survival in the placebo group of 28.2 months was extended to 42.7 months in the NP001 treatment group and to 45.5 months in patients with low creatinine levels.

Long-term effects of NP001 treatment were suggested by previously reported detailed biomarker analyses [22,23]. Two biomarker analysis studies of specimens available from the phase 2A study were performed. In the initial phase 2A study, patients aged ≤ 65 years and CRP > 1.13mg/L showed resolution of microbial translocation (MT) [22,23] that also responded to NP001 clinically as measured by VC and ALSFRS-R changes. Plasma levels of nine factors were significantly different after 6 months in the trial as compared to placebo. These included the end-of-study values related to wound healing and immune regulatory factors (Interleukin 10 (IL-10), epidermal growth factor (EGF), transforming growth factor beta 1 (TGF-β1), neopterin] and lowering of 5 associated with inflammation (lipopolysaccharide (LPS), LPS binding protein (LBP), soluble CD163 (sCD163), interleukin 18 (IL-18), hepatocyte growth factor (HGF)). Four factors were not abnormal to begin with and did not change (interferon gamma (IFN-ɣ), tumor necrosis factor-alpha (TNF-α), interleukin 6 (IL-6), interleukin 8 (IL-8)). Importantly, all biomarker evaluations were performed on plasma from patients one month after their final dose of NP001. The persistent changes in factors, such as EGF, HGF, and TGFB1, all of which have plasma half-lives of <2 min, are consistent with a long-lasting immune regulatory effect mediated by NP001 chlorite.

Historically, serum creatinine levels at the time of ALS presentation predict the rate of disease progression [24,25]. Low values reflect the loss of muscle mass secondary to inflammatory changes involving the neuromuscular interface. Chio et al. showed that ALS patients presenting with below the normal low-limit serum creatinine value cutoff point have significantly more rapid rates of disease progression [24]. In a meta-analysis of survival from studies of over 14,000 ALS patients, the creatinine level at the time of diagnosis was associated with survival, with high (normal) levels surviving the longest [25]. Rates of creatinine decline correlated with other ALS outcome measures, with the highest rates being associated with the worst survival [26]. A detailed evaluation of these markers showed that lower than the median level of creatinine and elevated CRP values were associated with the most rapidly progressive subset of ALS [24,25].

### 4.2. NP001 Is a Regulator of Innate Immune Activation

Results presented in the current study show that NP001 treatment conveys a survival benefit in ALS patients, showing the greatest VC response. ALS patients with lower serum levels of creatinine had VC loss slowed by close to 50% and a survival benefit of almost a year and a half after a six-month intermittent administration of NP001. Figure 6A shows a simple schematic of the innate immune self-regulatory cycle [22]. Innate immune activation creates inflammatory granulocytes and macrophages, and at the same time, the acute phase response makes CRP, which begins shutting off the inflammatory drive. The byproduct of the oxidative burst process, hypochlorous acid (HOCl), is neutralized by free taurine carried within these cells to make taurine chloramine (TauCl). TauCl is a dominant regulator of NFkB, which causes macrophages to become suppressive and feedback on the inflammatory cells to complete the regulatory cycle. This cycle is ongoing everywhere in the body under very tight regulation. Figure 6B takes into account the innate immune system recognition of TDP43 in the NMJ, triggering a blood-derived innate immune acute phase reaction. In this case, whether aging plays a role or the inflammatory signal overwhelms the feedback system, there are insufficient regulatory signals from the suppressive macrophages to control tissue damage with the consumption of muscle, both skeletal and diaphragmatic, leading to respiratory failure and death. Figure 6C shows a schematic of how NP001 (NaClO_2_) acting as a prodrug supplies the innate immune self-regulatory cycle an extra boost of TauCl, the inflammation regulatory product deficient in a chronic inflammatory setting like that of ALS. NaClO_2_ is converted to HOCl at the site of inflammation, catalyzed by heme-associated iron in the myeloperoxidase (MPO) complex [27]; the heme element in MPO is oxidized by chlorite, killing the MPO activity and ceasing inflammation [28]. The release of TGFB1 by the alpha 2 macroglobulin (A2M) dimer adds another powerful suppressor of inflammation to the system [22]. In addition, the dimeric form of A2M is highly efficient in the clearance of misfolded proteins and assists in the removal of the immune-activating misfolded proteins such as TDP43, further slowing the pathogenic inflammatory drive [22].

### 4.3. NP001 Extension of Life Is Novel and Unique among ALS Drugs

The OS study from Forrest et al. [15] showed a significant survival extension in the majority of patients evaluated in those 65 years old and under. All of the recent natural history studies of those at highest risk for developing ALS implicate chronic activation of the innate immune system, specifically in the group of individuals 65 years old and younger. As demonstrated by Forrest, patients over 65 years old progressed more rapidly in the OS study, with both the NP001 and placebo arms showing faster rates of decline, as has been reported in the literature for this group [13]. The current study provides evidence for a clinically less heterogeneous population of ALS patients, one in which the innate immune activation at the peripheral NMJ plays a critical role in the evolution of disease. This is the population with evidence for innate immune activation (plasma CRP > 1.13 mg/L), reversible with NP001. By comparison, Riluzole is reported to increase survival for patients with stage 4 disease only for 2–3 months [29], but it is the first oral drug approved for ALS currently in use by 65% of patients. There is no clear evidence that Riluzole changes measures of ALSFRS-R or VC in patients taking it as compared to placebo. Edaravone, approved for ALS as an IV drug in 2017 and was only approved in the US and Japan, affected approximately 8% of ALS patients, and none of the follow-up clinical trials could confirm efficacy [30]. An oral form of the drug was approved in 2022; however, a phase 3 trial showed no significant activity [31]. Qalsody is a molecular targeted drug that was developed by Biogen, and the target patient population is 1–2% of ALS patients with SOD1 mutations. It showed no activity in the pivotal trial at 6 months but did cause a measure of neurofilament light chain (NfL) to decrease. This biomarker-related decrease was central to this drug’s approval for only the 1% of ALS patients with a rapidly progressive form of familial ALS having a SOD1 genetic mutation [32]. No comparable data on any drug in ALS patients have shown the survival extension observed in the NP001 trials [15].

## 5. Conclusions

This is the first study to provide evidence that ALS is an immunologic disease. NP001 chlorite corrects one thing, and that is the innate immune activation cycle that normally keeps the body from over- or under-reacting to factors such as misfolded proteins. The overactivation of this system starts in the periphery at the NMJ and spreads into the central nervous system as a secondary process. This report also confirms that muscle loss, as measured by evaluation of serum creatinine levels and VC, is linked to the degree of immune activation, and the most abnormal at baseline do the best after immune correction with NP001. During the NP001 ALS 6-month trials, patients with characteristics of those with innate immune activation (elevated inflammation markers such as CRP and neutrophils in patients ≤ 65 years of age) showed significantly better outcomes (changes in ALSFRS-R and VC scores) than placebo. Most importantly, the current study reports, for the first time, that the diseased innate immune system can be put back into balance, allowing, in the case of ALS, almost a year and a half in extended survival after just six months of intermittent therapy. Future studies with NP001 having shown efficacy in a subset of patients with innate immune dysfunction will expand into other subsets, wherein drugs may have selective effects. NP001, after approval, will explore additional higher doses and optimize the IV regimen while searching for an oral form that is safe and effective, including the possibility of using a delivery system that bypasses the acid pH of the stomach.

## 6. Limitations of the Study

New data relating serum creatinine levels to VC and overall survival (OS) were presented from two combined NP001 phase 2 clinical trials. Although these were all post hoc analyses, all patients who had completed six-month trials were included in the overall completer analyses. While the survival data reported from the OS study show a median survival extension in the low creatinine NP001-treated arm from 28.4 to 45.5 months, NP001 treatment stopped at 6 months. Because these trials ended in 2012 and 2017, it is unlikely that ALS patients received any other drug than riluzole; however, the true post-trial events are unknown and could have potentially confounded the OS analysis. Although a phase 3 trial with prospectively defined endpoints related to those included in this report would confirm beyond a doubt the efficacy of NP001, the difficulty of following participants for at least 3–4 years to confirm survival makes this effort difficult and costly from both time and ethical perspectives.

## Figures and Tables

**Figure 1 biomedicines-12-02362-f001:**
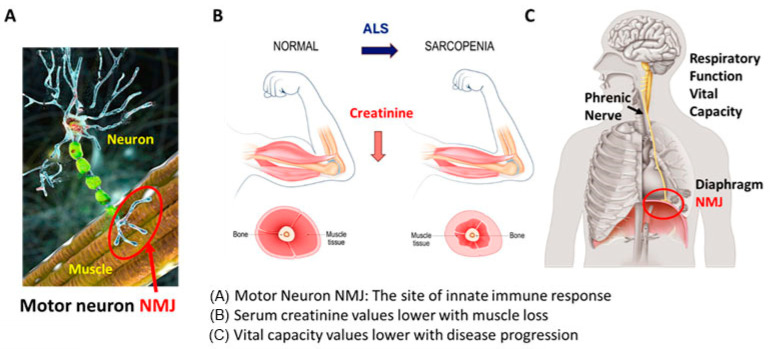
Innate immune destruction of the neuromuscular junction (NMJ) leads to loss of muscle and respiratory function and death. (**A**) The motor neuron axon outgrowths interact with the muscle at the neuromuscular junction (NMJ, red circle) (images licensed from istock). This is the site where inappropriately processed disease-associated proteins such as TDP43 are recognized by the innate immune system and begin the process of motor neuron and muscular dysfunction. (**B**) Loss of NMJ function leads to muscle loss (images licensed from Shutterstock). In addition, the acute phase reaction consumes muscle as a source of rare amino acids, such as phenylalanine, tryptophan, and tyrosine, turning normal muscle into a wasted sarcopenic muscle. Blood creatinine levels quantify the muscle as a source of APP amino acids, and CRP levels increase. (**C**) The diaphragm is innervated by the phrenic nerve, an outgrowth of a lower motor neuron (images licensed from Shutterstock). VC measurements indirectly measure the function of the NMJ.

**Figure 2 biomedicines-12-02362-f002:**
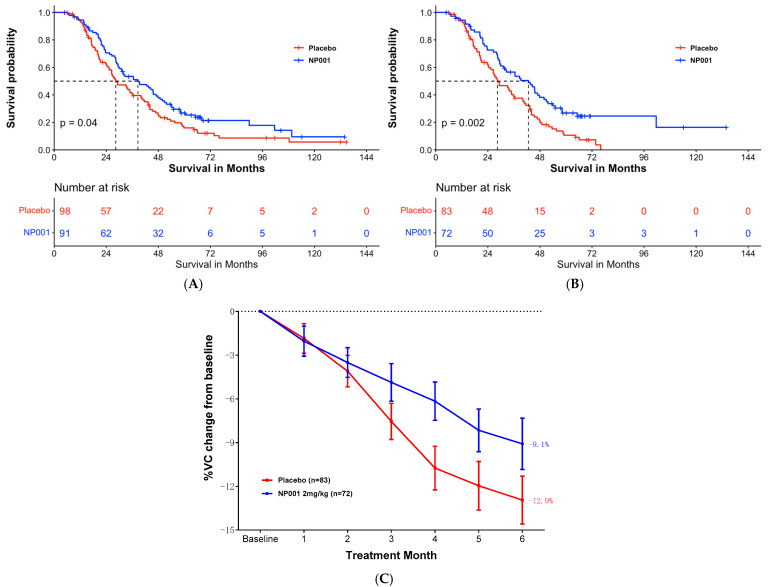
NP001 efficacy defined by survival analysis and % VC change in phase 2A and 2B completers. (**A**) Kaplan–Meier curve of survival probability for patients who received NP001 at a dose of 2 mg/kg compared with placebo in completers. In the completers, the median survival (95% confidence interval (CI)) over the entire follow-up duration was 38.6 months (95% CI: 29.9, 47.8) and 28.4 months (95% CI: 25.4, 40.0) in the 2 mg/kg NP001 (blue) and placebo (red) groups, respectively (log-rank, *p* = 0.04). Patients treated with NP001 had better survival than those on placebo: hazard ratio (HR) = 0.72 (95% CI: 0.52, 0.99). (**B**) Kaplan–Meier curve of survival probability for patients who received NP001 at a dose of 2 mg/kg compared with placebo in completers with CRP > 1.13 mg/L at baseline. In the completers with baseline CRP > 1.13 mg/L, the median survival (95% confidence interval (CI)) over the entire follow-up duration was 42.7 months (95% CI: 31.3, 50.9) and 28.4 months (95% CI: 24.4, 40.0) in the 2 mg/kg NP001 (blue) and placebo (red) groups, respectively (log-rank, *p* = 0.002). Patients treated with NP001 had a significant benefit of survival than those on placebo: hazard ratio (HR) = 0.57 (95% CI: 0.39, 0.82). (**C**) Change in % of VC from baseline over 6 months in participants on NP001 compared with placebo in completers with high CRP at baseline. Percentage VC change from baseline for participants treated with NP001 (*n* = 72, blue) is compared with the placebo group (*n* = 83, red). Bars represent mean of % VC change from baseline ± SEM. Average %VC lost over the 6 months of study: NP001: −9.1% (−1.5% per month); placebo: −12.9% (−2.2% per month). The NP001 treatment arm lost 29% less respiratory function than the placebo arm by the end of the study (Wilcoxon test, *p* = 0.02).

**Figure 3 biomedicines-12-02362-f003:**
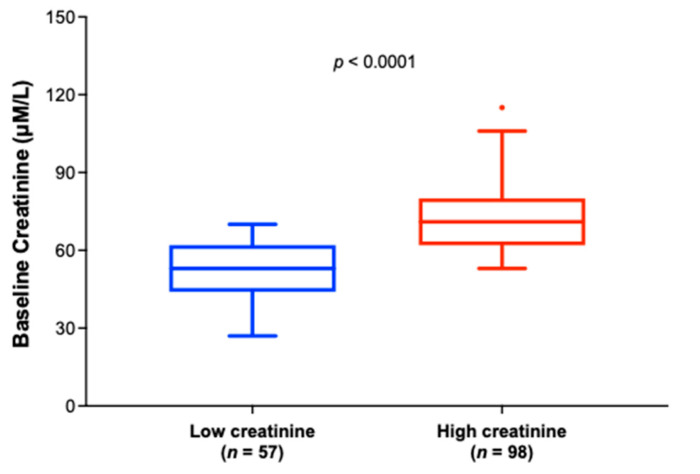
Box and whisker plots depicting the distribution of baseline serum creatinine values for the low creatinine group (baseline creatinine < 71 µM/L for males or <53 µM/L for females) (*n* = 57, in blue) and the high creatinine group (baseline creatinine ≥ 71 µM/L for males or ≥ 53 µM/L for females) (*n* = 98, in red). Results show that the median baseline creatinine value was statistically significantly lower in the low creatinine group compared with the high creatinine group (Wilcoxon test, *p* < 0.0001).

**Figure 4 biomedicines-12-02362-f004:**
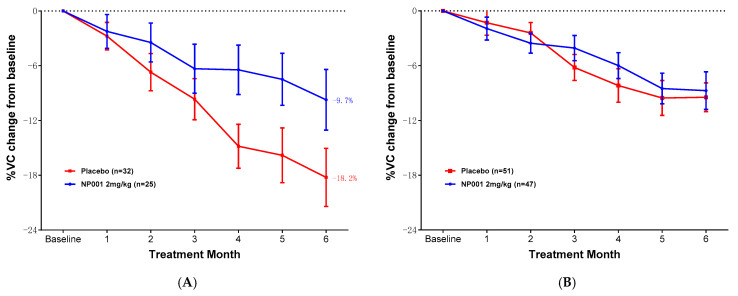
NP001 efficacy defined by % VC change by creatinine groups in phase 2A and 2B completers with baseline plasma CRP levels > 1.13 mg/L. (**A**) Change in % of VC from baseline over 6 months in participants on NP001 compared with placebo in those with low creatinine at baseline in completers with CRP > 1.13 mg/L. Percentage VC change from baseline for participants treated with NP001 (*n* = 25, blue) compared with the placebo group (*n* = 32, red). Bars represent mean of % VC change from baseline ± SEM. Average %VC lost over the 6 months of study: NP001: −9.7% (average −1.6% per month); placebo: −18.2% (average −3.0% per month). The NP001 treatment arm lost 46% less respiratory function than the placebo arm by the end of the study (Wilcoxon test, *p* = 0.02). (**B**) Percentage change from baseline in VC over 6 months in participants on NP001 compared with placebo in those with high creatinine patients with CRP > 1.13 mg/L at baseline. Mean change from baseline in percent VC for participants treated with NP001 (*n* = 47, blue) compared with the placebo group (*n* = 51, red). Bars represent mean of % VC change from baseline ± SEM. There was no significant difference between NP001-treated patients and placebos by the end of the study (NP001 = −8.7% vs. placebo = −9.5%) (Wilcoxon test, *p* = 0.30).

**Figure 5 biomedicines-12-02362-f005:**
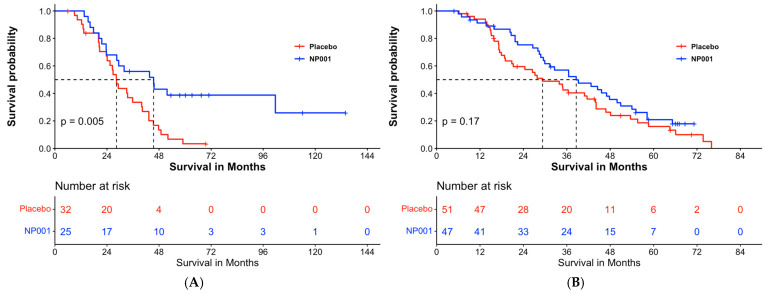
Overall survival by low and high creatinine groups in phase 2A and 2B completers with baseline plasma CRP levels > 1.13 mg/L. (**A**) Kaplan–Meier curve of survival probability for patients who received NP001 at a dose of 2 mg/kg (blue) compared with placebo (red) in the low creatinine group. The median survival (95% confidence interval (CI)) over the entire follow-up duration among those with low serum creatinine at baseline was 45.5 months (95% CI: 28.5, NA) and 28.4 months (95% CI: 24.1, 40.3) in the 2 mg/kg NP001 and placebo groups, respectively (log-rank, *p* = 0.005). The associated hazard ratio (HR) was 0.41 (95% CI: 0.22, 0.78). (**B**) Kaplan–Meier curve of survival probability for patients who received NP001 at a dose of 2 mg/kg compared with placebo in the high creatinine group. The median survival (95% confidence interval (CI)) over the entire follow-up duration was 38.6 months (95% CI: 29.9, 50.9) in the NP001 treatment and 29.3 months (95% CI: 20.8, 44.1) in the placebo group (log-tank, *p* = 0.17) with the associated HR 0.73 (95% CI: 0.46, 1.1).

**Figure 6 biomedicines-12-02362-f006:**
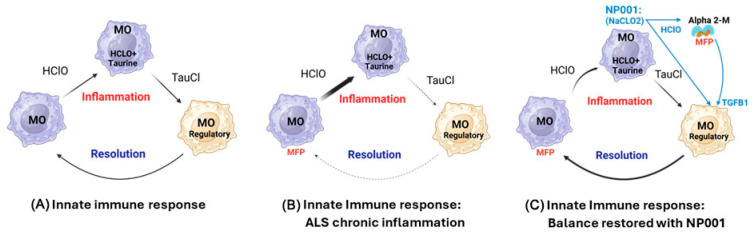
ALS is a systemic innate immune disease: NP001 resets the self-regulatory cycle in ALS patients, mitigating loss of muscle function (images licensed from BioRender). The innate immune system is balanced between activating inflammatory signals and resolving signals after the process initiating the inflammation has been resolved. (**A**) Inflammation yields the oxidative molecule HOCl, which is rapidly neutralized within macrophages (MO) by free taurine, creating chloramine (TauCl). TauCl reverses inflammation, and the resulting phagocytic wound healing cell makes factors that shut down inflammation. (**B**) In ALS, misfolded proteins (MFPs) such as TDP43 are recognized at the neuromuscular junction by cells in the peripheral immune system, causing inflammation. In ALS patients, there is insufficient production of regulatory molecules, allowing inflammation to persist and damaging the NMJ, leading to both neuronal and muscular damage to progress unabated. (**C**) NP001 is converted by inflammatory cells into HOCl, then TauCl, providing an augmentation to the TauCl pathway. In addition, excess HOCl causes dimerization of alpha-2 macroglobulin (Alpha 2-M), creating a high-affinity clearance structure for misfolded proteins such as TDP43. The dimer releases a preformed TGFB1, a known regulator of NFkB, to enhance the resolution process [22]. Re-establishment of innate immune balance slows the loss of muscle and vital capacity, leading to the prolongation of survival.

**Table 1 biomedicines-12-02362-t001:** Baseline demographics and characteristics of phase 2A and 2B completers ^1^ with plasma CRP > 1.13 mg/L at baseline.

	NP001 2 mg/kg	Placebo	
Characteristics	(*n* = 72)	(*n* = 83)	*p*-Value
Sex, *n* (%)			0.49
Female	24 (33.3%)	23 (27.7%)	
Male	48 (66.7%)	60 (72.3%)	
Age at baseline, year	56.3 ± 10.6	55.6 ± 10.4	0.49
Site of ALS onset, *n* (%)			
Bulbar	9 (12.5%)	14 (16.9%)	0.50
Limb	63 (87.5%)	69 (83.3%)	
El Escorial criteria for ALS, *n* (%)			NS
Definite	32 (44.4%)	35 (42.2%)	
Possible	6 (8.3%)	6 (7.2%)	
Probable	29 (40.3%)	36 (43.4%)	
Probable laboratory supported	5 (6.9%)	6 (7.2%)	
Concurrent riluzole use, *n* (%)			1.0
Yes	54 (75.0%)	62 (74.7%)	
No	18 (25.0%)	21 (25.3%)	
ALSFRS-R score at baseline, mean ± SD	38.4 ± 4.6	37.4 ± 5.4	0.32
Vital capacity at baseline, mean ± SD	93.3 ± 19.4	89.8 ± 18.3	0.37
Months since ALS symptom onset ^2^, mean ± SD	19.62 ± 8.45	18.11 ± 8.08	0.25
Creatinine at baseline (µM/L), mean ± SD	65.0 ± 16.6	65.0 ± 15.4	0.75

Abbreviations: n, number of participants. NS, not significant. SD, standard deviation. ^1^ Completers, the participants who had completed the 6-cycle treatments and had ALSFRS-R total score assessment at the end of study. ^2^ Months from ALS symptom onset to baseline.

**Table 2 biomedicines-12-02362-t002:** Baseline demographics and characteristics of participants defined as low and high baseline creatinine ^1^ in phase 2A and 2B completers ^2^ with baseline plasma CRP > 1.13 mg/L.

	Low Creatinine	High Creatinine	
Characteristics	(*n* = 57)	(*n* = 98)	*p*-Value
Sex, *n* (%)			0.001
Female	8 (14.0%)	39 (39.8%)	
Male	49 (86.0%)	59 (60.2%)	
Age at baseline, year	52.5 ± 11.2	57.9 ± 9.5	0.003
Site of ALS onset, *n* (%)			0.16
Bulbar	5 (8.8%)	18 (18.4%)	
Limb	52 (91.2%)	80 (81.6%)	
El Escorial criteria for ALS, *n* (%)			NS
Definite	22 (38.6%)	45 (45.9%)	
Possible	5 (8.8%)	7 (7.1%)	
Probable	26 (45.6%)	39 (39.8%)	
Probable laboratory supported	4 (7.0%)	7 (7.1%)	
Concurrent riluzole use, *n* (%)			0.85
Yes	42 (73.7%)	74 (75.5%)	
No	15 (26.3%)	24 (24.5%)	
ALSFRS-R score at baseline, mean ± SD	36.5 ± 5.2	38.7 ± 4.8	0.008
Vital capacity at baseline, mean ± SD	93.6 ± 21.1	90.2 ± 17.4	0.58
Months since ALS symptom onset ^3^, mean ± SD	21.24 ± 8.08	17.39 ± 8.07	0.004
Creatinine at baseline (µM/L), mean ± SD	52.2 ± 8.9	72.5 ± 14.3	< 0.0001

Abbreviations: n, number of participants. NS, not significant. SD, standard deviation. ^1^ Low and high creatine groups: baseline creatinine levels for males < 71 µM/L and females < 53 µM/L were defined as low creatinine group, and high creatinine group were males ≥ 71 µM/L and females ≥ 53 µM/L of creatinine values at baseline. ^2^ Completers: participants who had completed the 6-cycle treatments and had ALSFRS-R total score assessment at the end of study. ^3^ Months from ALS symptom onset to baseline.

**Table 3 biomedicines-12-02362-t003:** Baseline demographics and characteristics of low creatinine group ^1^ treated with NP001 vs. placebo in completers ^2^ with baseline plasma CRP > 1.13 mg/L.

	NP001 2 mg/kg	Placebo	
Characteristics	(*n* = 25)	(*n* = 32)	*p*-Value
Sex, *n* (%)			0.44
Female	2 (8.0%)	6 (18.8%)	
Male	23 (92.0%)	26 (81.3%)	
Age at baseline, year	52.2 ± 14.2	52.8 ± 8.3	0.80
Site of ALS onset, *n* (%)			0.64
Bulbar	3 (12.0%)	2 (6.3%)	
Limb	22 (88.0%)	30 (93.8%)	
El Escorial criteria for ALS, *n* (%)			NS
Definite	11 (44.0%)	11 (34.4%)	
Possible	1 (4.0%)	4 (12.5%)	
Probable	12 (48.0%)	14 (43.8%)	
Probable laboratory supported	1 (4.0%)	3 (9.4%)	
Concurrent riluzole use, *n* (%)			0.77
Yes	19 (76.0%)	23 (71.9%)	
No	6 (24.0%)	9 (28.1%)	
ALSFRS-R score at baseline, mean ± SD	37.0 ± 4.6	36.2 ± 5.8	0.80
Vital capacity at baseline, mean ± SD	98.4 ± 22.6	89.9 ± 19.4	0.16
Months since ALS symptom onset ^3^, mean ± SD	22.23 ± 8.12	20.47 ± 8.10	0.44
Creatinine at baseline (µM/L), mean ± SD	53.8 ± 9.6	51.1 ± 8.3	0.21

Abbreviations: n, number of participants. NS, not significant. SD, standard deviation. ^1^ Low creatinine group: participants with baseline creatinine levels for males < 71 µM/L and females < 53 µM/L. ^2^ Completers: participants who had completed the 6-cycle treatments and had ALSFRS-R total score assessment at the end of study. ^3^ Months from ALS symptom onset to baseline.

## Data Availability

The data are available through Neuvivo, Inc. upon request.

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
