# Peer review of "Systemic Innate Immune System Restoration as a Therapeutic Approach for Neurodegenerative Disease: Effects of NP001 on Amyotrophic Lateral Sclerosis (ALS) Progression"

_biomedicines, 2024, doi:10.3390/biomedicines12102362_

Round 1
Reviewer 1 Report (Previous Reviewer 2)
Comments and Suggestions for Authors
The authors studied the effects of NP001 on amyotrophic lateral sclerosis progression. Generally, the experiments were well-designed, and the results were interesting. It is recommended to accept this manuscript after revisions.
Q1, Figure 6 is not clear enough to see the words
Q2, better illustrate the pros and cons for different drugs on treatment of amyotrophic lateral sclerosis progression
Q3, better show the direction for future research.
Q4, is it possible to use nano carriers to deliver the drug for improving the therapeutic efficacy?
Author Response
The authors studied the effects of NP001 on amyotrophic lateral sclerosis progression. Generally, the experiments were well-designed, and the results were interesting. It is recommended to accept this manuscript after revisions.
Q1, Figure 6 is not clear enough to see the words
Figure 6 has been modified as suggested.
Q2, better illustrate the pros and cons for different drugs on treatment of amyotrophic lateral sclerosis progression
As the reviewer suggested, we have included the pros and cons of different drugs for the treatment of amyotrophic lateral sclerosis progression in the last paragraph of “Discussion” which were highlighted in yellow. A new reference #32 has also been added.
Q3, better show the direction for future research.
The direction of future research has been added in the last part of “Conclusion” and highlighted in yellow.
Q4, is it possible to use nano carriers to deliver the drug for improving the therapeutic efficacy
It will be challenging, at least for now.
Reviewer 2 Report (Previous Reviewer 1)
Comments and Suggestions for Authors
In this resubmitted manuscript, the authors have addressed all my previous comments adequately. I only think that there is no need to divide the introduction into subsections.
Author Response
In this resubmitted manuscript, the authors have addressed all my previous comments adequately. I only think that there is no need to divide the introduction into subsections.
The subtitles in the “Introduction” have been removed in the revised manuscript, as the reviewer suggested.
This manuscript is a resubmission of an earlier submission. The following is a list of the peer review reports and author responses from that submission.
Round 1
Reviewer 1 Report
Comments and Suggestions for Authors
The manuscript discusses the therapeutic potential of NP001 in treating Amyotrophic Lateral Sclerosis (ALS). The study pooled data from two six-month phase 2 clinical trials, showing that NP001 extends survival in ALS patients by restoring balance in the innate immune system. Notably, the greatest benefits were observed in patients with significant muscle loss. The findings support the role of systemic innate immune activation in ALS progression and suggest NP001 as a promising disease-modifying treatment.
Some modifications are required:
1. The full term "Amyotrophic Lateral Sclerosis" should be used in the title rather than the abbreviation "ALS".
2. In the introduction, a paragraph the defines the background of NP001 is required.
3. What is exactly "ALSFRS-R total score assessment"? An explanation is required.
4. The p values should be added to all the tables.
5. In the discussion, all the abbreviated biomarkers should be fully defined (full terms).
6. A concise conclusion is required.
Comments on the Quality of English LanguageMinor editing is required.
Reviewer 2 Report
Comments and Suggestions for Authors
The authors studied on the effects of NP001 for ALS progression. Generally speaking, the experiments were well designed and the results are interesting. It is suggested to accept this manuscript after revisions.
Q1, better offer more comparison for different drugs to treat the same disease. pls illustrate pros and cons of each drug.
Q2, is it possible to use imaging to real time evaluate the therapeutic effects?
Q3, There are some typos, please double check, like umol/L
Q4, the style of figures should be unified, like in figure 2, there are different types